# Molecular Mechanisms and Health Benefits of Ghrelin: A Narrative Review

**DOI:** 10.3390/nu14194191

**Published:** 2022-10-08

**Authors:** Zheng-Tong Jiao, Qi Luo

**Affiliations:** 1Queen Mary School, Medical College, Nanchang University, Nanchang 330006, China; 2Department of Histology and Embryology, Medical College, Nanchang University, Nanchang 330006, China

**Keywords:** ghrelin, growth hormone secretagogue receptor, neurohumoral regulation, homeostasis

## Abstract

Ghrelin, an endogenous brain–gut peptide, is secreted in large quantities, mainly from the stomach, in humans and rodents. It can perform the biological function of activating the growth hormone secretagogue receptor (GHSR). Since its discovery in 1999, ample research has focused on promoting its effects on the human appetite and pleasure–reward eating. Extensive, in-depth studies have shown that ghrelin is widely secreted and distributed in tissues. Its role in neurohumoral regulation, such as metabolic homeostasis, inflammation, cardiovascular regulation, anxiety and depression, and advanced cancer cachexia, has attracted increasing attention. However, the effects and regulatory mechanisms of ghrelin on obesity, gastrointestinal (GI) inflammation, cardiovascular disease, stress regulation, cachexia treatment, and the prognosis of advanced cancer have not been fully summarized. This review summarizes ghrelin’s numerous effects in participating in a variety of biochemical pathways and the clinical significance of ghrelin in the regulation of the homeostasis of organisms. In addition, potential mechanisms are also introduced.

## 1. Introduction

Ghrelin, which is a type of peptide hormone composed of 28 amino acids, is mainly expressed in the stomach, with a small amount being expressed in the hypothalamus, pituitary gland, or peripheral organs. It was first discovered in 1999. To investigate the effect of gastric tissue extracts on GHSR, Chinese hamster ovarian (CHO) cell expression of rat GHSR (CHO-ghsr62) was constructed. The results show that the gastric tissue extract could increase the intracellular calcium concentration of CHO-ghsr62 cells. Pigment chromatography revealed that ghrelin, a novel hormone, activated the GHSR [1]. The artificially encoded ghrelin mRNA was hybridized with the mRNA transcribed by gastric fundus cells. The results show that the hybridization signal of oxyntic cells was intense, which confirms that ghrelin is secreted richly in the oxyntic cells of the gastric fundus. However, the hybridization signal was weak between pyloric gland cells and upper small intestinal cells, which indicated that ghrelin secretion is rare in pyloric gland cells and upper small intestinal cells [2]. It is synthesized and secreted by the X/A-like cells of the oxyntic gastric glands in rodents, whereas ghrelin is secreted by P/D1 cells in humans [1,2]. Two structures of ghrelin are synthesized in the gastrointestinal (GI) tract, o-n-octanoylated ghrelin (acylated ghrelin, AG) and des-acylated ghrelin (unacylated ghrelin, UAG). It has been found that UAG does not perform any biological activity against GHS receptors [3]. During ghrelin synthesis, serine 3 residues are o-n-octanoylated [1]. UAG in the stomach and blood is more abundant than AG [3]. Ghrelin is formed from a 117-amino acid prohormone (preproghrelin) through post-translation. Except for AG, UAG and obestatin are also produced [4]. Obestatin was discovered and isolated in 2005 and can antagonize the appetite-stimulating effect of ghrelin [4]. However, the endogenous receptor of obestatin is controversial. Initial studies have shown that the receptor for obestatin is an orphan G protein-coupled receptor-GPR39 [4], and later studies have confirmed this point of view [5]. However, it has been suggested that zinc ions enhance the GPR39 downstream signal intensity, a process that cannot be observed in the presence of obestatin, which suggests that the endogenous ligand of GPR39 is the zinc ion rather than obestatin. It has been demonstrated that the obestatin receptor may vary between different tissues [6,7,8]. Some have also hypothesized that the specificity of obestatin in appetite suppression is caused by binding it to the glucagon-like peptide-1 receptor (GLP-1R) [9]. The enzyme that catalyzes UAG to AG is ghrelin-o-acyltransferase (GOAT) [10,11]. GOAT belongs to a group called membrane-bound o-acyltransferase (MBOAT4), which is responsible for the octoacylation of ghrelin. GOAT mRNA is most abundant in the stomach, but a small amount can be detected in the intestine. In the other organs, only the testis can reveal a certain amount of GOAT mRNA [10]. O-n-octoacylation is necessary for its binding to ghrelin’s specific receptor GHSR1a [1]. GHSR1a is a 277-amino acid G-protein-coupled receptor expressed abundantly in the gastric, hypothalamus, and pituitary glands [12,13]. GHSR1a is also expressed in immune cells [12] and in peripheral organs such as the pancreatic islet, adrenal gland, thyroid gland, lungs, liver, and kidney [14,15]. Ghrelin can regulate GH synthesis and secretion by activation of PKC, PKA, MAP kinase, and transcription of Pit-1 transcription factor gene fragments in growth-hormone-secreting cells [16]. Ghrelin also participates in various signaling pathways and helps produce different physiological responses (Figure 1).

Not only is Ghrelin a growth hormone secretagogue, but it also participates in various metabolic regulation activities in the body. As a ubiquitous brain–gut peptide hormone, it plays a multi-dimensional role in regulating the body’s metabolic balance, inflammatory response, reward feeding, depression and anxiety behavior, and cardiac function. Studies have found that ghrelin plays a protective role in damaged tissues or organs, mainly by stimulating the secretion of other hormone-secreting axes, activating anti-inflammatory factors, and inhibiting the production of inflammatory factors [17,18,19,20]. While ghrelin is still in early stages of development, a considerable amount of academic research has accumulated in this field, which suggests that the multi-dimensional impacts of ghrelin are complex. A comprehensive review of the relevant literature is especially helpful in synthesizing the key research insights and unveiling major research trends in this field. In this review, we aim to expound on the physiological effects of ghrelin and a pre-clinical experiment.

## 2. Ghrelin and the GI Tract

Many studies have shown that ghrelin is secreted in various parts of the GI tract (e.g., duodenum, jejunum, colon). Ghrelin secretion correlates with feeding status. Studies have shown that ghrelin secretion reaches its peak before food intake and gradually decreases to normal levels within an hour after food intake [21]. It also significantly relates to the acceleration of gastric emptying [21,22,23]. In addition, ghrelin has a certain promoting effect on the growth and development of GI organs, and this effect is significantly correlated with the age of mice. For example, ghrelin significantly delayed the development of the stomach and pancreas in neonatal mice before 7 weeks of age. However, in mice older than 7 weeks, ghrelin significantly increased the cell proliferation in the stomach and pancreas. This significant promoting effect may be related to endogenous insulin-like growth factor-1 (IGF-1) and expression in mice [24,25,26].

In addition to research on ghrelin’s effect on feeding and organ development, many studies have shown that exogenous ghrelin plays an important role in protecting against pathological changes of organs and tissues. The healing effect of ghrelin on gastric ulcers was confirmed in many studies. This effect may be related to the repair of the mucosal epithelium, the increase in mucosal epithelial blood flow, and the inhibition of pro-inflammatory factors. Ghrelin exhibited protective and therapeutic effects on different tissues, including those of the heart [27], kidney [28], spinal cord [29], pancreas [30,31,32,33], stomach and duodenum [34,35], colon [36,37,38], and oral mucosa [39,40].

A certain correlation has been found between gastric and duodenal ulcers and circulating ghrelin levels. In gastric ulcers caused by H. pylori, ghrelin expression significantly decreased, along with appetite and BMI, which were improved after the healing of gastric ulcers [41,42]. Ghrelin could significantly reverse ethanol-induced gastric ulcers and mucosal epithelial bleeding, and had a significant effect on the reduction of the gastric ulcer area. Cyclooxyganese-2 COX-2 activity is necessary for gastric ulcer healing [43]. In addition, ghrelin-mediated gastric ulcer repair may be related to endogenous nitric oxide (NO) release, vagus nerve stimulation, and sensory nerve stimulation [44]. In the treatment of indomethacin-induced gastric tissue injury, ghrelin could significantly reduce the degree of gastric mucosal epithelial injury and the ulcer area, and significantly inhibit the expression of pro-inflammatory factors such as TNF-α and IL-β. This process was significantly correlated with the level of heme oxygenase-1 (HO-1) [45]. This anti-inflammatory and antioxidant effect of ghrelin also showed a good preventive effect in a drug-induced colitis model in mice [36,37,38]. In other models of gastric ulcers induced by acetic acid, researchers also compared the GH, IGF-1 levels, and the degree of gastric ulcer healing after the injection of exogenous ghrelin and IGF-1 in mice with and without pituitary resection, and reached the same conclusion: the degree and rate of gastric ulcer healing were both improved after exogenous ghrelin injection, and this effect could be mediated by the release of endogenous GH and IGF-1 [34]. It is possible that, for this reason, ghrelin did not show such an anti-inflammatory effect on duodenal ulcers in mice with hypophysectomy [25]. Downregulation of ghrelin expression was also observed in mice with gastric outlet obstruction (GOO), a pathological state that is similar to a gastric ulcer in clinical settings [46]. In summary, ghrelin can repair and protect against gastric and duodenal injury, but more molecular and pre-clinical studies on the specific mechanism should be discussed and implemented to confirm this finding. Such an anti-inflammatory effect was also reflected in the improvement of oral mucositis by ghrelin.

Oral mucositis is ulcerative inflammation that occurs in the mouth, also known as an oral ulcer. Ghrelin plays a significant role in the alleviation of this disease by reducing the production of the pro-inflammatory factor IL-β and increasing the proliferation of mucosal cells. Moreover, it could effectively reverse the harm caused by surgery in mice who had undergone salivary gland resection [39]. Furthermore, this effect was shown to be related to the secretion of endogenous GH and IGF-1 [40].

In addition, ghrelin has been shown to have protective and therapeutic effects on acute pancreatitis. Previous studies showed that ghrelin inhibited the production of TNF-α and reversed the pancreatic enlargement induced by cerulein. Ghrelin upregulates the GHSR-1a and ghrelin protein expression, which has been speculated to be involved in reducing acute pancreatitis. This protective pathway was shown to be dependent on capsaicin-sensitive sensory nerves [47,48].

## 3. Ghrelin and Orexis

### 3.1. Ghrelin’s Involvement in the Neuroregulation of Appetite

Ghrelin stimulates GHSR, which was officially named the ghrelin receptor in 2006, on the cells of the ventromedial hypothalamus nucleus (VMH) [49]. The binding of ghrelin to GHSR1a activates the phospholipase C signaling pathway, which results in an increase in inositol phosphate and the activation of protein kinase C (PKC), followed by the release of Ca2+ from the intracellular store and an increased intracellular calcium concentration [50]. The activation of GHSR1a also inhibits K+ channels, which allows for Ca2+ entry through voltage-gated L-shaped channels [51,52]. The increase in the intracellular calcium ion concentration can activate CaMKK2 kinase in the downstream pathway and induce AMPK phosphorylation. CaMKK2 forms a stable complex with AMPK that enables subsequent regulation of metabolism and response to ghrelin effects [53]. The activation of AMPK also requires the involvement of Sirt1 signaling. It was found that Sirt1 could activate AgRP neuron activity to promote the appetite and participate in the inhibition of proopiomelanocortin (POMC) neuron activity, which is involved in appetite suppression [54]. Thus, the initiation of the ghrelin signaling cascade requires co-activation of the p53-Sirt1 and CaMKK2 pathways to promote AMPK phosphorylation, which inhibits ACC and also promotes the release of carnitine palmitoyltransferase 1 (CPT1) [55]. Two subtypes of CPT1, CPT1A and CPT1C, are involved in this signaling pathway. CPT1A is a rate-limiting enzyme that acts on the outer mitochondrial membrane to regulate β-oxidation and thereby control fatty acid metabolism. Fatty acid metabolism increases mitochondrial ROS expression, and upregulation of this expression induces UCP2 expression. CPT1C in the neuronal endoplasmic reticulum was also shown to be activated by ghrelin, which increased the expression of ceramide and thereby promoted appetite and upregulating the expression of NPY/AgRP [56,57]. Activation of AMPK increased CPT1 expression in the hypothalamus. Injection of the CPT1 inhibitor reduced UCP2 mRNA [58], which suggests that ghrelin-induced UCP2 mRNA expression is mediated by the AMPK-CPT1 pathway. Phosphorylation of AMPK inhibits acetyl-CoA carboxylase (ACC), decreases melonyl-CoA production, and relieves the inhibition of CPT1A and CPT1C. The production of UCP2 and ceramide are transmitted to the NPY/AgRP neurons of the hypothalamic arcuate nucleus (ARC) through an unclear mechanism, where they promote the transcription of NPY and AgRP genes by activating transcription factors. Current studies have shown that appetite is regulated by synaptic transmission of signals to ARC, which stimulates transcription factors such as CREB and FOXO1, which then promote the transcription of Npy and AgRP genes [59]. The activation of POMC neurons reduces appetite, while the activation of NPY/AgRP neurons plays an opposite role. These opposite functions are based on their target neurons, which aid in the expression of the melanocorticoid 4 receptor (MC4R). POMC activates MC4R-expressing neurons in the paraventricular nucleus (PVH) of the hypothalamus, thereby reducing appetite. By contrast, NPY/AgRP neurons antagonized these effects [60,61]. These neurons send efferent signals to key hypothalamic circuits, including the production of neuropeptide Y (NPY), AgRP, and POMC. NPY/AgRP neurons contain the inhibitory neurotransmitter GABA, which projects to POMC neurons and directly inhibits the activity of POMC neurons [62]. In summary, ghrelin’s neural conduction in different regions of the hypothalamus affects the appetite, indirectly affects food intake, and participates in the regulation of metabolic disorders under pathological conditions (Figure 2).

### 3.2. Ghrelin in Obesity

The specific mechanisms and signaling pathways of ghrelin in the regulation of obesity are unclear. It is believed that ghrelin increases blood glucose and appetite, thus causing obesity, and inhibiting the activity of ghrelin or its receptor attenuates obesity, reduces blood glucose, and promotes fat metabolism [63]. Ghrelin secretion is sensitive to the concentration of blood glucose under hypoglycemia. Previous studies showed a significant increase in the plasma concentrations of ghrelin in glucagon receptor knockout (Gcgr−/−) mice [64]. Moreover, long-term fasting leads to an increased serum ghrelin level. After abour an hour after eating, the circulating ghrelin gradually decreases. In addition, blocking the interaction between ghrelin and/or ghrelin receptors aggravated hypoglycemia symptoms [65], which suggests that ghrelin may be a good combination agent for patients who do not respond well to glucagon receptor antagonists. When glucagon receptor antagonists are used for treating type 1 diabetes, plasma ghrelin concentrations increase to antagonize drug-induced hypoglycemia. Thus, ghrelin can reverse hypoglycemia when used in the treatment of hypoglycemia or in combination with diabetic drugs, and has great therapeutic significance in alleviating the side effects of antidiabetic drugs.

Prader–Willi syndrome, known as hypotonia–mental retardation gonadal retardation–obesity syndrome, is a rare genetic disease. After infancy, the rate of hyper appetite, obesity, learning disabilities, and short temper increase [66]. Studies have shown that circulating ghrelin levels are significantly associated with pathological obesity [67,68]. However, whether the pathologic obesity caused by gene mutations is associated with the mutation or deletion of ghrelin-related genes is uncertain. The occurrence of obesity may be related to the destruction of hypothalamic neural circuits and the conduction disorder of neurotransmitters. Patients with Prader–Willi syndrome showed hyperfunction in the brain regions related to appetite and pleasurable eating, such as the basal amygdala, hypothalamus, and hippocampus [69]. Previous studies have mainly focused on ghrelin receptor inhibitors and GOAT inhibitors, which affect the synthesis and function of ghrelin. However, inhibitors related to these substances have not shown significant effects in the disease’s treatment [70]. Combining these findings with the abnormal function of neurotransmitters and neurons involved in the reward brain circuits, we hypothesized that ghrelin establishes abnormal connections in the hypothalamus to promote appetite. Inhibiting neuron activity or knocking out neuron-related transmitter genes may reverse Prader–Willi syndrome-induced pathological obesity. The related signaling pathways are worthy of further investigation. There may be a new metabolic disorder that is associated with this pathologic obesity.

Metaflammation, a new concept that correlates obesity with inflammation, is caused essentially by the dysfunction of the immune metabolism, and the pressure of nutrition and energy leads to the appearance of signaling pathways and cascades without severe immune response symptoms. It is similar to a chronic immune response and can exist for a long time [71]. This kind of metabolic inflammation can damage the micro-structure of organs and cause abnormal physiology. Inflammatory cytokines such as TNF inflammatory mediators, toll-like receptors, and their ligands have been implicated in obesity-induced inflammatory responses in adipocytes [72]. These inflammatory factors also participate in the progression of obesity-associated diseases [73,74]. This means that pathological obesity leads not only to weight gain but also to disorders in the human immune system, triggering immune responses and causing chronic inflammation. Ghrelin is involved not only in the regulation of energy intake but also in the adjustment of the immune response by inflammatory factors [17]. In fact, metaflammation sets up a bridge that links the two aspects together.

LEAP2 is a liver-produced antimicrobial peptide. The first LEAP2 variant was isolated from a human hemofiltration peptide library [75]. It was first identified to be a specific ghrelin receptor antagonist in 2018. LEAP2 could completely antagonize GHSR, the ghrelin receptor, and this inhibition could not be overcome by increasing the ghrelin concentration. The expression pattern of LEAP2 is the opposite to that of ghrelin. The ghrelin concentration reaches its peak within 24 h of fasting, when the concentration of LEAP2 is the lowest. The concentration of ghrelin gradually decreases, while the concentration of LEAP2 gradually increases after 1 h of eating [76]. Bariatric surgery, such as gastric bypass and vertical cannulation gastrectomy (VSG), has been identified to reduce the LEAP2 level [18,77]. LEAP2 is a powerful GHSR antagonist that incapacitates AG-induced activation of NPY neurons. It also acts as a GHSR inverse agonist that incapacitates the constitutive activity of GHSR [77]. A large volume of data show that human obesity is associated with higher plasma LEAP2 and lower plasma AG, similar to obesity in mice. This means that obese people have a higher plasma LEAP2/AG molar ratio. In addition, plasma LEAP2 was found to be positively correlated with some adverse metabolic parameters related to obesity, including BMI, body fat percentage, HOMA-IR, fasting blood glucose and serum triglyceride, VAT volume, and IHCL content [78,79,80]. However, a recent study of childhood obesity showed that plasma LEAP2 levels in overweight/obese children appeared to contradict previous findings [81]. In this research, plasma LEAP2 levels decreased in overweight/obese children, along with UAG, and the ratio of AG to UAG decreased. The mechanism for the difference is unknown. It is speculated that the growth and development of ghrelin receptors or other endocrine organs in children are different from those in adults.

In addition to the effects of GHSR antagonist LEAP2 on metabolic regulation, inhibitors of GOAT, a transferase that catalyzes ghrelin production, have also been investigated recently. A synthetic GOAT inhibitor, GO-CoA-TAt, was discovered in 2010, which reduced circulating concentrations of AG and attenuated weight gain in mice that were fed a high-fat diet [82]. During subsequent experiments, it was found that the plasma AG concentration was significantly reduced at 6 h after treatment with the GOAT inhibitor compared with a saline treatment and there was no significant change in ghrelin levels during the subsequent period [83]. However, the specific mechanisms of ghrelin O-acyltransferase inhibitors need to be further investigated because it is important for research on how to treat obesity caused by pathological metabolic abnormalities, as well as for the development of new targets for weight loss drugs.

### 3.3. Ghrelin and Anorexia Nervosa

Anorexia nervosa (AN) refers to an eating disorder characterized by individuals intentionally reaching and maintaining a significantly lower weight than normal through dieting and other means. Patients with this disorder have a variety of neuroendocrine abnormalities, as well as hormonal or neuropeptide dysregulation related to appetite and satiety [84,85]. As ghrelin is a hormone that regulates appetite, the circulating ghrelin level appears to be significantly correlated with certain indicators in patients with AN, as evidenced by the first pre-clinical research on the topic, which revealed that ghrelin levels in AN patients were significantly elevated [86].

Because of the potent appetitive effects of ghrelin, some studies have considered the use of exogenous ghrelin in the treatment of AN patients. It was reported that ghrelin injection could significantly improve anorexia in an activity-based anorexia (ABA) mouse model, although there was no significant increase in body weight [87]. However, it was also pointed out that in patients with AN, the appetitive effects of ghrelin might not exist independently, and the treatment of patients with AN with exogenous ghrelin slightly increased their appetite [88]. A study also monitored growth hormone, prolactin, and cortisol levels after ghrelin infusion and found that these hormones did not increase significantly, but drowsiness occurred [88]. It is speculated that the reason may be the weight of the patient; as the weight of the patient increases, the appetitive effect of ghrelin becomes more significant, which has been confirmed in some studies [89]. Other studies have used exogenous ghrelin in AN patients and found no significant increase in GH level, which indicates that exogenous ghrelin’s influence on the poor efficacy of the treatment of patients with AN may be due to the abnormality of the AN patient’s own appetite regulating pathways, which mainly manifests in the symptom of ghrelin resistance [90]. The current clinical research on the relationship between ghrelin and AN is based on a small number of experimental groups, and it is possible that a larger volume of clinical research will emerge in the future to support or refute the currently unclear view.

## 4. Ghrelin in the Cardiovascular System

Ghrelin also participates in regulating various aspects of the cardiovascular system. Previous studies have shown that ghrelin mRNA is found in both peripheral vessels and cardiomyocytes [1], and GHSR1a is found in vascular smooth muscle cells and cardiomyocytes [15]. Subsequent studies have shown that ghrelin is involved in the regulation of the cardiovascular system and is associated with obesity. Intravenous ghrelin reduces substantially systolic blood pressure (SBP) in obese people, presumably by stimulating muscle sympathetic nerve activity (MSNA) through an unknown mechanism [91]. The presence of ghrelin receptors in preganglionic sympathetic vasoconstrictor nerve fibers but not postganglionic motor nerve fibers suggests that ghrelin acts as an inhibitory ligand to inhibit the vasoconstrictive action of preganglionic sympathetic vasoconstrictor nerve fibers, thereby reducing blood pressure [19]. Ghrelin performs a cardiovascular function by significantly reducing peripheral vascular resistance, which induces an immediate decrease in blood pressure and leads to a compensatory increase in MSNA [92,93]. However, the exact mechanism of ghrelin in the regulation of sympathetic neurons is still unclear. In addition to its role in the regulation of cardiovascular activity by controlling peripheral vasodilation or contraction and peripheral resistance, ghrelin has a positive effect in improving chronic heart failure (CHF), myocardial infarction, post-infarction myocardial fibrosis, ventricular remodeling, and the reversal of arrhythmias [92,94].

Cardiac cachexia is a wasting disease mainly related to CHF. For patients with CHF, the presence of cachexia is the main reason for their poor condition and prognosis. Ghrelin levels have been shown to be elevated in patients with CHF and cachexia compared to those without cachexia [94]. Subcutaneous injection of ghrelin in CHF rats with cachexia inhibited left ventricular enlargement and shortening and attenuated ventricular remodeling due to CHF [95].

In addition to improving cardiac dysfunction in rats after myocardial infarction, ghrelin treatments have been found to produce fewer inflammatory cytokines and activate fewer microglia than CHF rats [96]. Clinical studies showed that in the treatment of cachexia, quinolones, the recent treatment of patients with chronic diseases including CHF, exploited similar signaling pathways in humans to that of ghrelin-GHSR, which provides a crucial clue in the study of ghrelin-GHSR-dependent therapeutic targets [97].

Ghrelin has also been found to have a positive effect on various CHF conditions, including myocardial ischemia, acute myocardial infarction, myocardial fibrosis, and ventricular remodeling. Thus, we can develop antagonists and agonists for the investigation of the ghrelin downstream signaling pathway and apply them to clinical treatment. Previous studies showed that intravenous injection ghrelin could significantly increase the heart rate and left ventricular systolic pressure (LVSP) of myocardial ischemia mice in the experiment of Langendorff heart perfusion in vitro. Left ventricular end-diastolic pressure (LVEDP) decreased in the ischemia/reperfusion (I/R) of mice [98]. These data suggest that ghrelin may improve cardiac dysfunction caused by myocardial ischemia. In addition, reperfusion of ghrelin increased the coronary blood flow, which might contribute to improved left ventricular function. However, the improvement of cardiac function might not be mediated by the traditional ghrelin-GHSR-dependent pathway [99], which indicates that there may be a new signaling pathway mediating the improvement of ventricular function. Recent studies showed that both UAG and AG could improve cardiac function by promoting cell survival and inhibiting myocardial cell inflammatory corpuscle and apoptosis. However, only AG could promote food intake and growth hormone secretion owing to the difference in the signal pathway [3,4].

In the model of doxorubicin-induced cardiomyopathy, the ventricular shortening rate and ejection fraction significantly decreased, the degree of myocardial fibrosis increased, and the activities of caspase-3 and other factors involved in apoptosis increased. UAG could improve these pathological changes [100,101]. In addition, ghrelin significantly inhibited the over-expression of TLR4, NLRP3, and caspase-1 protein induced by ischemia-reperfusion (I/R) injury. These results indicate that ghrelin can reduce oxidative stress and inflammatory response in myocardial I/R injury. Myocardial ischemia and necrosis can be improved by inhibiting the TLR4/NLR3 signaling pathway [102] (Figure 3).

Subcutaneous injection of UAG significantly improved systemic hemodynamics and significantly reduced myocardial infarct size [103]. In addition, intravenous administration of AZP-513, a UAG analogue, significantly reduced myocardial infarct size during early reperfusion by inhibiting the Bax/Caspase 3-induced apoptosis pathway, ROS accumulation, and oxidative stress-induced necrosis [104,105,106]. Recent studies have shown that epithelial–mesenchymal transition (EMT), where cardiomyocytes transform into cardiac fibroblasts, is involved in cardiac diseases after myocardial ischemia. Ghrelin inhibits fibroblast markers such as αSMA, Slug, and Snail, increases the expression of epithelial cell markers such as CD31 and E-cadherin, and upregulates Smad7 to inhibit the TGF-β/Smad2/3 fibrogenic pathway by activating the GHSR/AMPK signaling. This effect, in turn, significantly improved myocyte fibrosis (MF), reduced collagen deposition in perivascular fibrotic areas, and improved myocardial infarction [107]. Ghrelin can also improve cardiomyocyte apoptosis caused by the surge of reactive oxygen species by stimulating the Raf-1-MEK1/2-ERK1/2-BAD signaling pathway in the left ventricular infarction area, activating antioxidant enzyme activity, and enhancing myocardial contractile force [108].

Remote ischemic preconditioning (RIPC), a method of exposing a tissue or organ to one or more cycles of nonfatal I/R, has been proven to be cardioprotective in the event of fatal acute I/R injury to the heart [109,110]. The concept of ischemic preconditioning was first proposed by Dr. Murry from the United States in 1986. It means that repeated, short, non-traumatic, and non-harmful ischemic preconditioning training is often carried out on the human body, which can stimulate the human immune system and produce and release endogenous protective substances such as adenosine, bradykinin, and nitric oxide. These are involved in the protection of the myocardium and mitigate and resist the damage caused by ischemia and hypoxia in humans over a longer period of time. They can effectively avoid the occurrence of cerebral infarction, sudden cardiac death, and other cardiovascular and cerebrovascular diseases [110]. Some studies have shown that the growth hormone family is involved in distal ischemic regulation through the Janus kinase (JAK)/signal sensor and activator of transcription (STAT) pathway [111]. RIPC and UAG-injection-dependent plasma-UAG levels were shown to protect against myocardial I/R injury by activating the JAK2/STAT3 pathway and increasing the expression of STATtyr705 [112,113]. This indicates that UAG may be an important humoral factor involved in the cardioprotection of RIPC.

Angiotensin (Ang) and ROS are involved in myocardial remodeling in MI rats. Nuclear factor-erythroid associated factor 2 (Nrf2), a regulator of antioxidant enzymes, has also been found to be negatively associated with oxidative stress in stress-overloaded hearts [107,108]. By contrast, ghrelin upregulates Nrf2 in a ghrelin-receptor-dependent way, thereby inhibiting the NADPH/ROS. This leads to post-infarction and Ang II-induced ECM remodeling and the inhibition of myocardial fibrosis in cardiac fibroblasts [114,115,116].

Organic heart injury, especially ventricular tachycardia complicated by structural heart disease, is usually an arrhythmia that can lead to ventricular fibrillation or other serious outcomes, including sudden death. Ghrelin treatment significantly increased the ventricular fibrillation threshold (VFT), shortened the action potential duration (APD) dispersion and APD alternation, and promoted vagus nerve activity. Ghrelin also significantly reversed the abnormal expression of tyrosine hydroxylase in the peri-infarction region [117].

Both UAG and AG play important roles in the pathogenesis of structural heart disease, such as myocardial infarction and myocardial fibrosis due to myocardial ischemia, as well as ventricular remodeling. However, the specific mechanisms and synergistic regulatory factors of peptide hormones such as UAG, which are independent of GHSR signaling, still need to be investigated further.

## 5. Ghrelin in Stress-Induced Anxiety

Ghrelin is secreted by parietal cells in the stomach, enters the circulation, crosses the blood–brain barrier, and regulates brain regions that control anxiety and stress [118]. In rodent models of acute and chronic calorie restriction, circulating ghrelin levels rose with anti-anxiety and antidepressant action. Depression model mice injected with exogenous ghrelin had higher circulating levels of ghrelin, and less anxiety-like behavior in mice was observed in an elevated plus maze and forced swim tests [20]. This indicates that ghrelin plays an important role in resisting stress behaviors and emotions like anxiety and depression. Many stress models have shown the positive effects of ghrelin on depression and anxiety. For example, ghrelin levels consistently increased in models of chronic social defeat stress [20]; ghrelin receptor levels and ghrelin secretion in the hippocampus increased in the mouse model of chronic unpredictable mild stress (CUMS) [119]. Long-term injection of peripheral ghrelin significantly alleviated CUMS-induced anxiety- and depression-like behaviors. The results, combined with a large number of data and meta-analyses, indicate that ghrelin can be used as a marker of stress [120].

Ghrelin can regulate various brain regions, such as BLA and the hippocampus [121]. However, under different stress responses, the specific mechanism, downstream pathways, and neural conduction of ghrelin involved in antidepressant and anti-anxiety behaviors are unclear. Chronic stress has been shown to increase the formation of circulating endogenous acylated ghrelin and fear memory, and to cause the loss of ghrelin-binding receptors in the amygdala and desensitize the activity of ghrelin receptors in the amygdala [121]. This indicates that the degree of fear and the persistence of fear memory are related to the level of endogenous AG. The degree of fear may have different effects on the regulation of ghrelin. Long-term fear memory formation and short-term acute fear stimulation have certain differences in the regulation of ghrelin in the body. Ghrelin receptor knockout mice relied on endogenous ghrelin and ghrelin receptor interaction to produce anti-anxiety and antidepressant effects when subjected to acute caloric restriction. However, chronic caloric restriction, with or without ghrelin receptor knockout, could induce anti-anxiety and anti-depression effects [122,123]. This indicates that in chronic caloric restriction, the body resists anxiety and depression in a GHSR-independent manner, which is different from the effect in acute caloric restriction.

PTSD is a kind of delayed-onset mental disorder, which is mainly manifested as the repeated and involuntary emergence of trauma-related situations or contents in the patient’s thoughts, memories, or dreams, and also can appear as severe tactile reactions. Long-term chronic fear memory formation is strongly associated with the formation of post-traumatic stress syndrome. Studies have shown that the formation of post-traumatic stress is closely related to the increase in the circulating ghrelin levels. Activation of ghrelin receptors of any cause or long-term exposure to ghrelin activation increases the formation of fear memories, suggesting that ghrelin can be detected as a biomarker in post-traumatic stress syndrome [124]. This process is mediated by the ghrelin/ghrelin receptor/growth hormone system rather than by traditional glucocorticoids or epinephrine [125,126]. However, whether AG circulation is involved in HPA-mediated PTSD is unclear.

Elevated ghrelin is meant to relieve anxiety and depressive symptoms while increasing appetite, which may be an adaptive state of the body to cope with stress [119]. Therefore, it can be extended to consider whether there is a certain relationship between pathological obesity and depression related to abnormal ghrelin regulation. In the brain, high levels of ghrelin are present in the dentate gyrus, BLA, hippocampus, and other brain regions, in addition to the hypothalamus, which acts on the control of appetite and feeding [127]. Notably, these ghrelin-high regions have been strongly associated with depression-like emotions and behaviors [128,129]. For example, hippocampal integrity was reduced under CUMS, while ghrelin receptor-null mice exhibited depressive-like behavior in the absence of stress, demonstrating that both GHSR knockout and CUMS have deleterious effects on neurogenesis and spine density in the dentate gyrus (DG). Ghrelin/GHSR signaling pathways maintained hippocampal integrity and have intrinsic neuroprotective effects [130]. Dysregulation and abnormal neural connectivity in the parts of the brain that control appetite could also contribute to obesity symptoms [131,132]. It has been speculated that the imbalance in the regulation of brain circuits involved in ghrelin may contribute not only to metabolic imbalances leading to obesity but also to chronic stress disorders. In addition to this, a study aimed to determine whether a high-fat diet is caused by hypothalamus ghrelin resistance or whether knockout ghrelin receptor genes lead to the loss of activity of ghrelin, compared with normal healthy people. None of the anxiety samples had an obvious effect on behavior and fear conditioning, but could significantly reduce the preference for sugar water [133]. Overall, the neuroregulatory functions involved in ghrelin may produce different regulatory mechanisms based on different individuals (obese or healthy) and different emergency stimuli (acute or chronic).

## 6. Ghrelin in Cachexia

Cachexia has been widely investigated as a significant complication of various cancers in current clinical research. Generally, cachexia is defined as a state of extreme malnutrition in a severe wasting disease state, that is, a manifestation of extreme emaciation due to some disorder of self-regulation or decreased appetite. However, there is no effective way to significantly improve this condition. Ghrelin is an important new research direction for the treatment of this severe wasting syndrome. The antagonists of appetite-stimulating factors such as ghrelin, which inhibits appetite-stimulating neurons such as Npy/AgRP in the hypothalamus, were found to result in a variety of cachexia symptoms, such as loss of appetite, weight loss, and muscle atrophy in advanced cancer patients [134].

TNF-α and IL-6 have been found to be significantly elevated and involved in appetite reduction in patients with advanced cancer [73]. The administration of antagonistic agents against cytokines significantly improved cachexia [74]. Ghrelin could antagonize inflammatory factors such as TNF-α and IL-6 to significantly improve the decreased appetite caused by inflammatory cytokines in a GHSR-dependent way. The ghrelin receptor is expressed on the cell membrane of T cells [12]. Its activation could downregulate the synthesis and release of inflammatory cytokines in T cells [135].

It has been found that serum ghrelin concentrations are closely associated with upper GI tumors. In a variety of upper GI tract cancer cases, ghrelin levels dropped significantly, which indicates that the ghrelin level may be the early biomarker of cancer risk in the upper GI tract [136]. Ghrelin infusion could significantly improve cachexia in patients with advanced cancer. It could significantly enhance the nutritional intake, ensure the supply of basic nutrients, and improve muscle mass [137]. In patients with chronic respiratory failure, ghrelin could significantly improve the loss of appetite and weight loss, significantly promote breathing ability, and improve the survival index of patients and the prognosis of cancer [138]. Studies showed that the risk index between the placebo group were the same as the oral ghrelin group in terms of prognosis, which indicates that ghrelin is safe in improving the poor prognosis of patients, and that there is no adverse reaction in the tolerance of patients [139]. However, it was found that adverse symptoms such as insulin resistance and muscle atrophy could occur after ghrelin infusion for GI malignant tumors [140].

Anamorelin is a ghrelin derivative synthesized by mimicking the N-terminal Ser3 active region of endogenous ghrelin, which can also specifically bind to the active binding site of ghrelin receptors and produce a series of ghrelin-like active functions. Moreover, anamorelin can even directly act on appetite-stimulating neurons in the brain. Specifically, it can activate the body’s anorexic function [141]. This finding may have important implications for improving the symptoms of anorexia caused by cachexia. In Japan, the ghrelin receptor agonist anamorelin was approved for the treatment of four cancers: non-small cell lung cancer, gastric cancer, pancreatic cancer, and colorectal cancer. The results show that the ghrelin receptor agonist alleviated weight gain, improved muscle strength, and increased appetite in patients with advanced cancer cachexia. The ghrelin receptor agonist has also been shown to exert a benign promoting effect in cancers such as GI neoplasms and non-small-cell lung cancer [142,143,144]. Moreover, anamorelin could improve the muscle atrophy or GI mucosal damage caused by chemotherapy, and could also significantly increase the appetite [145]. These results indicate that anamorelin may be used in combination with other chemotherapeutic drugs to reduce the adverse reactions caused by chemotherapy and improve the negative emotions of patients after chemotherapy.

## 7. Conclusions

This review illustrated that ghrelin is not only orexigenic but also a factor with multiple regulatory functions. Obesity, digestive system disorders, cardiovascular disease, chronic inflammation, stress and anxiety, and cachexia caused by metabolic disorders are closely related to the regulatory function of ghrelin. However, owing to the lack of relevant studies and clinical data on ghrelin, the specific mechanism by which ghrelin regulates homeostasis remains unclear. There are controversies between the clinical conclusions of ghrelin and metabolic disorders, mainly owing to the small sample size, the existence of individual differences, and other reasons.

It is well known that metabolic homeostasis in the human body is jointly regulated by a variety of hormones. The hypothalamic–pituitary axis involved in the orexigenic regulation of ghrelin is also affected by a variety of hormones. For example, obestatin, which is produced by post-translational modification of preproghrelin, has a synergistic effect with ghrelin. Leptin is also synthesized and secreted by gastric fundus cells, which has a certain antagonistic effect with ghrelin. In contrast, recent studies have shown that mRNA transcription of ghrelin and its receptor is also detectable in islet B cells [4,52,72,135]. These results may be conducive to the combination of ghrelin with insulin, obestatin, leptin, and other hormones. The hypothalamic–pituitary gland secretion axis involved in each hormone deserves further in-depth analysis.

A more comprehensive analysis of the multi-disease regulation involved in ghrelin is warranted; for example, the epithelial–mesenchymal transition process that occurs in cardiomyocytes during cellular fibrosis caused by cardiac ischemia also occurs in tumor cell proliferation and metastasis, which is effectively ameliorated by ghrelin [134,136,137]. This important finding will facilitate the future comprehensive analysis and research of the ghrelin–GHSR signaling pathway and its downstream cascade in the development and spread of various cancers.

In addition to the regulation of AG, research on UAG is extremely active. Unlike AG, which has a known receptor, UAG is involved in the regulation of various systems in the body, but its receptor has not been clearly defined, and the research on the macroscopic changes of individual performance of UAG can still be deepened. It is particularly important to study the specific mechanism of UAG in regulating body homeostasis.

In conclusion, the molecular mechanisms regulating ghrelin synthesis and secretion need to be further studied. It is necessary to comprehensively explore the mechanism of ghrelin in a variety of diseases. The study of the specific targets of ghrelin and its downstream pathways will be the main direction of subsequent research. A comprehensive study of the relationship between ghrelin and various diseases and their complications will be of great significance for the screening of drug targets for related diseases and the development of new drugs (Figure 4).

## Figures and Tables

**Figure 1 nutrients-14-04191-f001:**
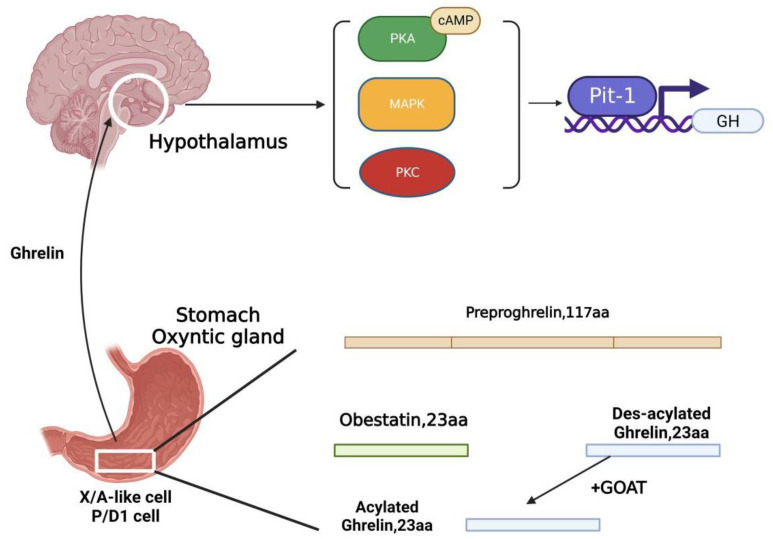
Ghrelin’s synthesis and actions through brain–gut transport. Ghrelin is synthesized from preproghrelin and transported through blood circulation to the hypothalamus, where it stimulates growth hormone secretion. PKA, protein kinase A; Camp, cyclic adenosine monophosphate; MAPK, mitogen-activated protein kinase; PKC, protein kinase C; GH, growth hormone; GOAT, ghrelin-o-acyltransferase. Created with BioRender.com.

**Figure 2 nutrients-14-04191-f002:**
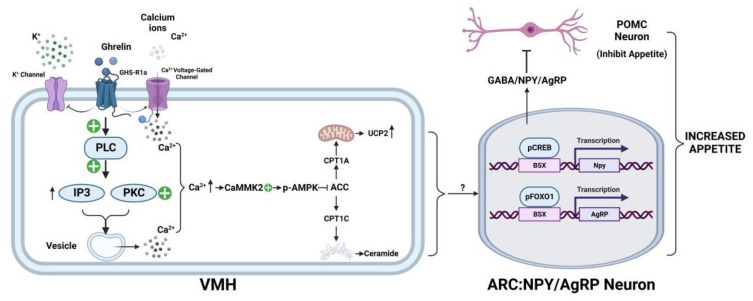
Ghrelin downstream signaling pathways in VMH and ARC. By activating intracellular downstream signaling molecules and promoting neurotransmitter transmission, ghrelin eventually promotes appetite. VMH, ventromedial nucleus of the hypothalamus; ARC, arcuate nucleus; PLC, phospholipase C; IP3, inositol triphosphate; PKC, protein kinase C; CaMKK2, calcium-dependent protein kinase kinase 2; MAPK, mitogen-activated protein kinase; CREB, cAMP-response element binding protein; FOXO1, Forkhead Box O1; BSX, brain-specific homeobox transcription factor; Npy, Neuropeptide Y; AgRP, agouti-related protein; GABA, γ-aminobutyric acid; POMC, pro-opiomelanocortin. Created with BioRender.com.

**Figure 3 nutrients-14-04191-f003:**
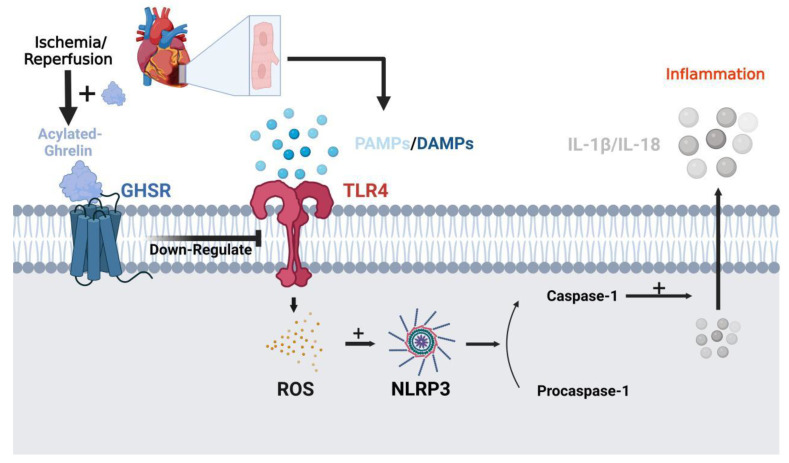
TLR4/NLRP3 signaling pathway inhibition by acylated-ghrelin to reduce I/R-induced cardiomyocyte infarction and inflammation. During myocardial I/R injury, TLR4 is expressed on the surface of the myocardial cell membrane, and PAMPs and DAMPs bind to it to release ROS. ROS activates NLRP3 inflammasome, which in turn promotes the activation of procaspase-1 to caspase-1. Caspase-1 promotes the release of inflammatory factors IL-1β and IL-18 from the cell and triggers the inflammatory response. Ghrelin can downregulate the expression of TLR4, thereby inhibiting the occurrence of the inflammatory response and alleviating myocardial injury caused by I/R. TLR4, Toll-like receptor 4; PAMPs, Pathogen-associated molecular patterns; DAMPs, damage associated molecular patterns; ROS, reactive oxygen species; NLRP3, NOD-like receptor thermal protein domain associated protein 3; IL-1β&IL-18, interleukin 1β&interleukin 18. Created with BioRender.com.

**Figure 4 nutrients-14-04191-f004:**
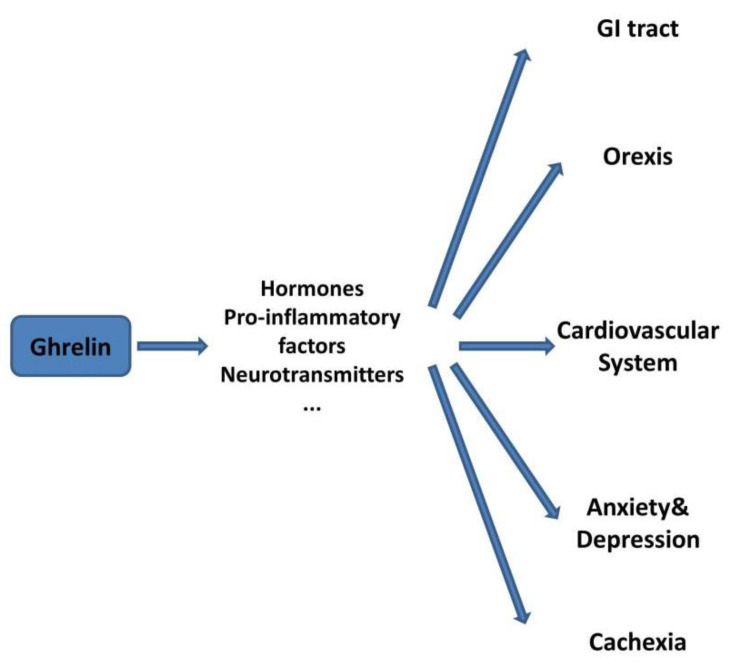
Ghrelin’s multifunctional role in different systems regulating homeostasis. Ghrelin is involved in the regulation of metabolic homeostasis, the digestive system, the cardiovascular system, emotional feedback, cachexia, and other aspects, and it is necessary to take all of these aspects into consideration. It is worth exploring whether these factors’ involvement in the downstream pathway regulated by ghrelin can link the various systems.

## Data Availability

Not applicable.

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
