# Peer review of "Molecular Mechanisms and Health Benefits of Ghrelin: A Narrative Review"

_nutrients, 2022, doi:10.3390/nu14194191_

Round 1
Reviewer 1 Report
1. The subject of this manuscript proposed for evaluation is extremely interesting. Unfortunately, the topic proposed by the authors in this review, as well as very well-described topic, have already been published in many magazines, including by MDPI. If this review applies for publication in such a reputable journal as Nutrients, it must meet a number of necessary conditions. The authors propose a quite logical arrangement of chapters and subsections. This is an extensive review, with a lot of literature (106 items). Unfortunately, in my opinion, the authors gave too little space to describe the role of ghrelin in inflammatory pathologies of the gastrointestinal tract (especially of the oral cavity, gastric and duodenal ulcers and pancreatitis) - both in the mechanisms of prevention and treatment of these inflammations. This requires a separate chapter or subsection. The authors' statement in the text that these topics are still in the introductory stages is somewhat outdated. I also draw your attention to a very extensive review papers from 2019 and 2022 (Int J Mol Sci. 2019 doi: 10.3390/ijms20071534 and doi: 10.3390/ijms221910571 Int J Mol Sci. 2021and Biomolecules 2022, https://doi.org/10.3390/biom12040517), which contains many more current reports on this topic - the authors should take this into account and update their references a bit. I am sure that these changes suggested to the authors will significantly increase the substantive value of this manuscript, which will obviously translate into a greater number of scientists interested in this topic.
2. The authors wrote: „Ghrelin is formed from a 117-amino acid pro-hormone (preproghrelin) through post-translation. Except for AG, UAG and obestatin are also produced [4].” This is definitely not enough. At this point, the authors should add information about obestatin and its action to the review article PMID: 25716961, especially the suggested obestatin receptors, and should at least mention the various effects of this related hormone.
3. In the text of manuscript, the authors did not describe and did not refer to experimental studies that concerned ghrelin. Also ghrelin, which activates the hormonal axis: growth hormone-IGF-1 and increases the release of endogenous IGF-1 (PMID:17033095), exhibits protective and therapeutic effects in different tissues, including the heart (PMID: 14556085), kidney (PMID: 16306169), spinal cord (PMID: 22949835), pancreas (PMID: 20814069, PMID: 21081804, PMID: 25594510, PMID: 17084100), stomach and duodenum (PMID: 22534700, PMID: 19439811), colon (PMID: 25838694, PMID: 27598133, PMID: 26769837, PMID: 26713317) and oral mucosa (PMID: 29151078, PMID: 24304579). It is necessary to refer to the articles. This should be added.
4. The authors in the “Introduction” section should describe the role of capsaicin-sensitive sensory serves in the protective effect of ghrelin (PMID: 28665321, PMID: 28468316).
5. Ghrelin has an influence on gastric and duodenal growth and expression of digestive enzymes in young mature rats (PMID: 17033095, PMID: 16337939) and on pancreatic development in young rats (PMID: 16391414). The authors should write about it.
6. Unfortunately, the authors, in my opinion too briefly described the physiological role of ghrelin and its role in the organism. Ghrelin signaling has been thoroughly investigated under conditions of anorexia nervosa ( https://doi.org/10.3390/ijms19072117).
7. The authors completely omitted the review in which there is a chapter on the role of ghrelin in the gastrointestinal tract, this should be added by the authors (Diverse and Complementary Effects of Ghrelin and Obestatin by Daniel Villarreal, Biomolecules 2022, https://doi.org/10.3390/biom12040517). Such a short extension of this topic will undoubtedly raise the quality of this manuscript.
Reviewer 2 Report
1. There are several areas where the use of English is less than precise. In the current version, English level negatively affects the quality of the manuscript. Professional English revision is necessary to improve the quality of the manuscript.
2. In the abstract, line 18-19, “physical functions” does not perfectly describe ghrelin’s effects described in the manuscript.
3. The introduction is a description of the synthesis and distribution of ghrelin. No general “introduction” on its health benefits, background and aim of the review article are provided.
4. There’s an overall problem with the abbreviations. All the abbreviation in the text should be consistent. Make sure to specify the abbreviation at first use only (e.g., Line 66: VMH, Line 69: protein kinase C, Line 103: POMC, etc.) and use it along the text.
5. Line 65: “Ghrelin involved..” should be corrected with “Ghrelin is involved..” or “Ghrelin involvement..”.
6. Paragraph 2.1., this is a very detailed paragraph on the ghrelin signaling pathways. There is lack of information on the effects of ghrelin at pre-clinical and clinical levels on appetite.
7. Paragraph 2.2., in this paragraph too much attention is given to the Prader-willi syndrome.
8. Line 121: “in regulation” should be modified with “in the regulation”.
9. A lot of typos should be addressed: (Line 205 researches-researchers; Line 216 the plural of person is people not persons; Line 221 Ghrelin performs-not perform; Line 402 complicaion; etc.).
10. Line 337, the sentence starting with “Antidepressant” and ending with “swim tests” is not clear.
11. Figure 4, in the figure legend, the sentence “In the future, different research directions on ghrelin should be studied due to multiple aspects.” should not belong to a figure legend. Authors should not simply state that different research directions should be studied due to multiple “aspects”.
Round 2
Reviewer 1 Report
The authors of the responses to the main comments, significantly improved the content of the manuscript, while increasing its scientific value.
Reviewer 2 Report
I am satisfied with the author's revisions. The current version of the manuscript has improved substantially in terms of quality.